# Control of oviductal fluid flow by the G-protein coupled receptor Adgrd1 is essential for murine embryo transit

Enrica Bianchi [1], Yi Sun [2,3], Alexandra Almansa-Ordonez[1], Michael Woods[4], David Goulding[5], Nadia Martinez-Martin[2] & Gavin J. Wright [1,6✉]

Dysfunction of embryo transport causes ectopic pregnancy which affects approximately 2% of conceptions in the US and Europe, and is the most common cause of pregnancy-related death in the first trimester. Embryo transit involves a valve-like tubal-locking phenomenon that temporarily arrests oocytes at the ampullary-isthmic junction (AIJ) where fertilisation occurs, but the mechanisms involved are unknown. Here we show that female mice lacking the orphan adhesion G-protein coupled receptor Adgrd1 are sterile because they do not relieve the AIJ restraining mechanism, inappropriately retaining embryos within the oviduct. Adgrd1 is expressed on the oviductal epithelium and the post-ovulatory attenuation of tubal fluid flow is dysregulated in *Adgrd1*-deficient mice. Using a large-scale extracellular protein interaction screen, we identified Plxdc2 as an activating ligand for Adgrd1 displayed on cumulus cells. Our findings demonstrate that regulating oviductal fluid flow by Adgrd1 controls embryo transit and we present a model where embryo arrest at the AIJ is due to the balance of abovarial ciliary action and the force of adovarial tubal fluid flow, and in wild-type oviducts, fluid flow is gradually attenuated through Adgrd1 activation to enable embryo release. Our findings provide important insights into the molecular mechanisms involved in embryo transport in mice.

[1] Cell Surface Signalling Laboratory, Wellcome Sanger Institute, Cambridge, UK. [2] Receptor Discovery Group, Microchemistry, Proteomics and Lipidomics Department, San Francisco, CA, USA. [3] Institute of Cardiovascular Sciences, University of Birmingham, Birmingham, UK. [4] Mouse Production Team, Wellcome Sanger Institute, Cambridge, UK. [5] Electron and Advanced Light Microscopy Suite, Wellcome Sanger Institute, Cambridge, UK. [6] Department of Biology, Hull York Medical School, York Biomedical Research Institute, University of York, Wentworth Way, York, UK. ✉email: gw2@sanger.ac.uk

Ectopic pregnancies occur when a fertilised egg implants and develops outside of the uterus, and in almost all cases, this occurs in the Fallopian tube resulting in a tubal pregnancy. The control of embryo movement through the oviduct is therefore thought to have an important role in this condition, but while generic risk factors that include previous IVF treatment and surgery have been identified, the underlying genetic causes and mechanisms are poorly characterised[1,2].

The oviduct promotes successful fertilisation by storing and supporting the capacitation of sperm, and providing a conduit for eggs and early embryos to reach the uterus. The transit of eggs and embryos through the oviduct is thought to be due to the combined action of several different factors which include the beating of cilia in an abovarial direction, periodic contractions of the surrounding muscle, and regulated secretions from the oviductal epithelium. One general feature of mammalian oviductal transport is that ovulated cumulus-oocyte-complexes initially move rapidly through the infundibulum to the ampulla, and are then halted for many hours at the ampullary-isthmic junction (AIJ) before continuing their journey to the uterus[3–7]. In humans, the AIJ is the site of fertilisation and the arrest of oocytes is likely to promote successful reproduction by providing a suitable environment for the gametes to meet and suppress polyspermy[8]. The pausing of tubal passage is a conserved feature of mammalian oviducts and has been described in several mammals including mouse[3,6], rabbits[7], horses[9], sheep, pig, guinea pig and cat[10]. None of the factors known to be involved in tubal transport provide a satisfactory explanation for this valve-like behaviour of the oviduct, and especially how it is unlocked to allow the developing embryos continued passage to the uterus.

Adhesion G-protein coupled receptors are a subfamily of 33 cell surface receptor proteins in humans that contain the archetypal seven transmembrane-spanning region that couples extracellular stimuli to intracellular signalling and appropriate cellular responses through G-protein activation. They are distinguished by the presence of an N-terminal extracellular region that varies considerably in length within the subfamily and contains protein domains whose known functions are to mediate cell and extracellular matrix interactions[11]. These receptors are expressed on a wide range of different cell types and gene-deficient mice have shown that they have a variety of functions ranging from immunoregulation[12,13], development of the nervous system[14], angiogenesis[15,16] and male fertility[17,18] although there have been no previous reports of a role in female reproduction. One characteristic feature is the presence of the GAIN (GPCR-Autoproteolysis INducing) domain which proteolytically cleaves the extracellular region at the GPS (GPCR Proteolytic Site) and reassociates non-covalently to form a heterodimeric receptor. Activation of these receptors is triggered by the binding of specific extracellular ligands which relieves an autoinhibitory region thereby releasing an activatory peptide[19,20]; however, many receptors are orphans, having no known ligand.

Here, we report on the action of a gene called Adhesion G-protein coupled receptor D1 (Adgrd1) which is expressed in the oviductal epithelium and female mice lacking this gene are sterile because embryos are retained within the ampulla. Our results demonstrate an essential role for Adgrd1 in embryo transport and because oviductal fluid flows are dysregulated in these animals, provides a plausible explanation for oviductal tubal locking and how it is regulated.

## Results

**Adgrd1-deficient female mice are sterile due to defective embryo transport**. To identify genes required for female fertility with unknown mechanisms of action, we interrogated the International Mouse Phenotyping Consortium database and identified a gene encoding an orphan member of the family of adhesion G-protein coupled receptors, Adgrd1[21]. We confirmed that female mice containing a targeted Adgrd1 gene-trap allele (Supplementary Fig. 1) were sterile, irrespective of the male genotype (Fig. 1a). Female Adgrd1-mutant reproductive tissues were morphologically normal, and ovulated the same number of eggs as wild-type littermates (Fig. 1b). Fertilisation of Adgrd1-deficient oocytes was unaffected both in vivo (Fig. 1c) and in vitro (Fig. 1d). We did not observe any embryo implantation in Adgrd1-deficient mothers (Fig. 1e), although $Adgrd1^{-/-}$ uteri could support embryo development (Fig. 1f, g) since transferred wild-type embryos developed normally until at least day 12.5 (Fig. 1g—inset). We consistently observed embryos denuded of cumulus cells that were ectopically located in the ampulla of mutant mothers (Fig. 1h, i), which had reached the expected morula stage by 2.5 dpc (Fig. 1j). The remnants of unfertilised eggs such as empty zonae pellucidae and dead oocytes were also frequently observed in the ampullae of mutant mothers (Fig. 1h). Together, these data demonstrate that $Adgrd1^{-/-}$ females are sterile because embryos are inappropriately retained in the ampulla, perhaps because they are unable to reverse the tubal lock that retains eggs at the AIJ.

**Adgrd1 is expressed in the oviductal epithelium**. To understand how loss of Adgrd1 regulates embryo passage through the AIJ, we first asked in which tissues Adgrd1 was expressed. Using the beta-galactosidase enzyme that is incorporated into the gene trap allele, we used whole mount X-gal staining to show that the Adgrd1 promoter was highly active in the isthmus in fertile Adgrd1 heterozygous mice (Fig. 2a, Supplementary Fig. 2a). The expression in the isthmus was clearer when the highly coiled oviduct was dissected away from the ovary and uterus and partially extended (Fig. 2b). To determine the location of the protein, we raised an antibody to the entire extracellular regions of mouse Adgrd1 and observed that the protein was expressed throughout the oviductal epithelial cells in the ampulla in both secretory and ciliated cells (Fig. 2c); the antibody did not stain oviductal epithelium from Adgrd1-deficient mice, demonstrating the specificity of the antibody (Fig. 2c). At higher magnification, the staining was concentrated at the cell surface in both ciliated and non-ciliated secretory cells (Fig. 2d). Staining sections of the isthmic region of the oviduct showed Adgrd1 was expressed in the epithelial cells that line the lumen of the oviduct and not the underlying smooth muscle (Fig. 2e). We did not observe any major changes in the level of transcription of Adgrd1 in either the ampullary or isthmic epithelium between the ovulatory period (0.5 dpc) and after the embryos had transited through the AIJ (1.5 dpc) (Supplementary Fig. 2b). These data show that Adgrd1 is expressed on the plasma membrane of both secretory and ciliated cells in the epithelium of the oviduct and is not regulated during the oestrous cycle.

**The structure and function of oviductal cilia and muscle are normal**. Embryo transport through the oviduct is thought to involve the concerted effects of oviductal fluid secretions, muscle contractions, and the action of cilia which beat in an abovarial direction towards the uterus, although their relative contributions have not been determined. To provide a mechanistic explanation for the lack of embryo transport, we first stained oviductal sections with a ciliary marker which revealed no overt differences in the ultrastructure of the epithelium nor the number or distribution of ciliated cells in the Adgrd1-deficient oviduct compared to heterozygous controls (Fig. 3a). Using electron microscopy, we similarly observed no differences in the ultrastructure of the oviductal cilia (Fig. 3b), nor the typical 9 + 2 arrangement of the

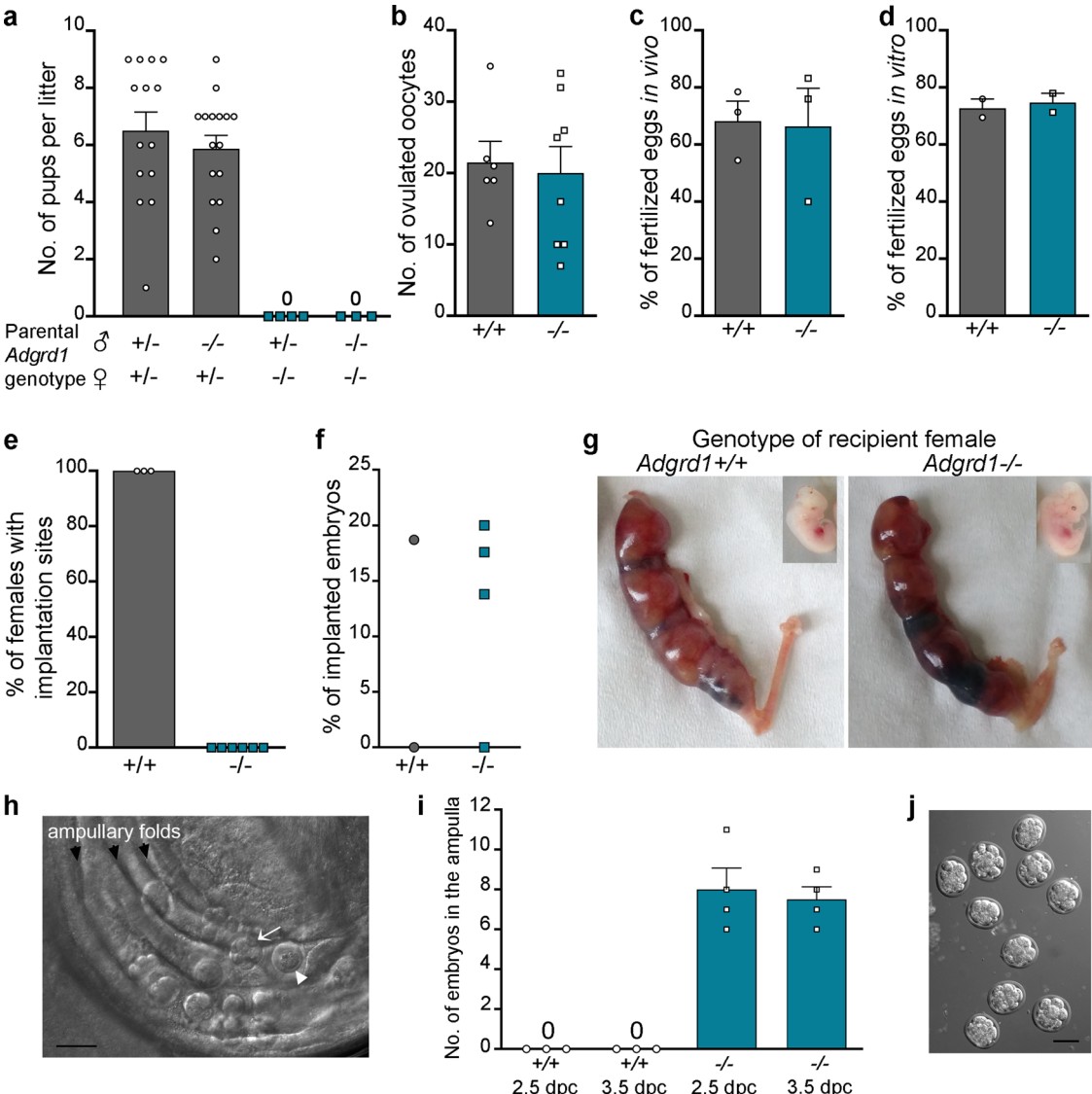

**Fig. 1 Adgrd1⁻/⁻ female mice are infertile due to defective embryo transport. a** *Adgrd1⁻/⁻* females are sterile irrespective of the male genotype. **b** *Adgrd1⁻/⁻* female mice ovulate comparable numbers of oocytes compared to wild-type controls. *Adgrd1⁻/⁻* oocytes are fertilised at normal frequencies both in vivo (**c**) and in vitro (**d**). **e** No implantation sites were observed in *Adgrd1⁻/⁻* females. **f** *Adgrd1* mutant uteri support implantation of transferred wild-type embryos. **g** Images showing implantation of transferred wild-type embryos in *Adgrd1⁻/⁻* and control uteri at 12.5 dpc; embryos developed normally (insets). **h** 2.5 dpc embryos are ectopically located within the ampulla of an *Adgrd1⁻/⁻* oviduct. Arrow identifies morula-stage embryo, arrowhead identifies a dead oocyte. Scale bar represents 100 μm. **i** Embryos are ectopically located in the ampulla of *Adgrd1⁻/⁻* mice at 2.5 dpc and 3.5 dpc. **j** Mutant *Adgrd1⁻/⁻* embryos recovered from the ampulla at 2.5 dpc have reached the morula stage. Scale bar represents 80 μm. Bars in **a**, **b**, **c**, **d**, and **i** represent mean ± SEM. Each data point in **a** represents a biologically independent mating pair. In **b**, **c**, **d**, **e**, **f** and **i** each data point represents a single mouse; source data are provided as a Source Data file. **h** and **j** are representative examples of at least five independent experiments.

microtubules (Fig. 3c). To investigate ciliary function more directly, we placed beads on explants of oviductal ampullary epithelium and quantified their velocity. Beads placed on both Adgrd1-mutant explants and controls moved in an aboarial direction (Supplementary Movie 1), and at similar speeds (Fig. 3d). These findings were consistent with the transport of the cumulus complexes from the infundibulum to the ampulla which was normal in *Adgrd1⁻/⁻* mice, and thought to be largely driven by ciliary action. We next examined the smooth muscle that lines the oviduct (the myosalpinx) and observed no overt defects in muscle organisation or structure in the mutant oviducts (Fig. 3e). We stained sections of the oviduct with a marker of smooth muscle and again observed no differences between wild type and

mutant (Fig. 3f, f′). Consistent with this, there were no differences in the thickness of the myosalpinx in either the ampullary or isthmic regions of the oviduct (Fig. 3g). By in situ observation of the oviduct, we did not detect any overt differences in the rhythm or spatial locations of muscle contractions in *Adgrd1*-deficient oviducts. To examine this in more detail, we placed beads into the lumen of an Adgrd1-mutant oviductal explant to determine the effects of muscle contractions on their movement and observed that they were not rigidly held in position, but rather were moved back and forth in pendulum-like movements due to the contractions of the surrounding muscle (Supplementary Movie 2). These observations were consistent with the view that muscle contractions mix rather than vectorially transport the contents of the

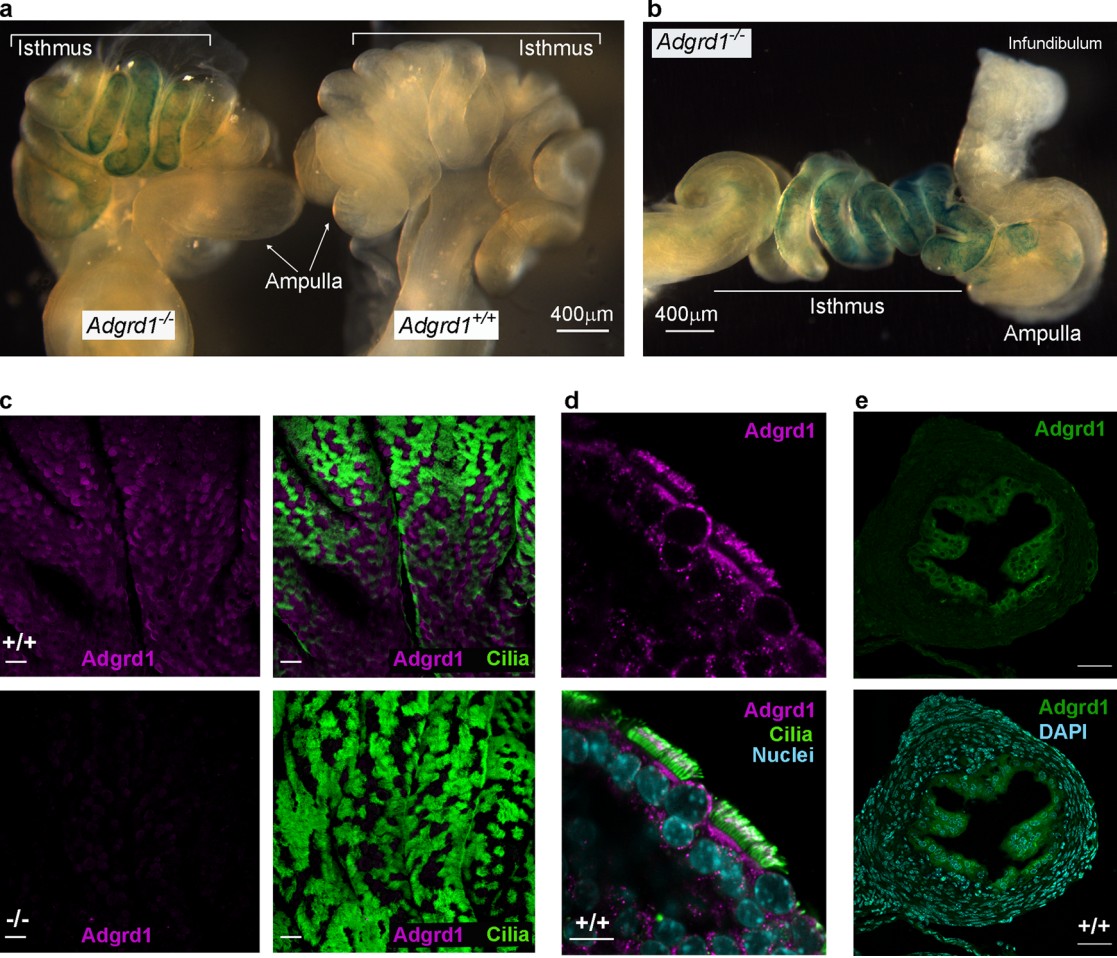

**Fig. 2 Adgrd1 is expressed on the plasma membrane of both secretory and ciliated cells in the epithelium of the oviduct. a** *Adgrd1* promoter is active in the isthmus of the oviduct as detected by whole mount X-gal staining using the *lacZ* reporter enzyme. Staining is detected in the homozygous oviducts (left) but not in the wild-type sibling control (right). **b** Whole mount X-gal staining of homozygous *Adgrd1* oviducts that have been removed from the ovary and uterus and partially extended to show *Adgrd1* promoter activity in the isthmus. **c** An Adgrd1 polyclonal antibody raised against the entire ectodomain specifically stains the ampullary epithelium (purple) from wild type (upper panels) but not *Adgrd1*-deficient mice lower panels); anti-acetylated tubulin staining was used to mark ciliated cells (green). **d** Adgrd1 is localised on the apical plasma membrane of ciliated and non-ciliated cells in the oviductal epithelium. Scale bar represents 10 μm. **e**, Expression of Adgrd1 in the epithelial cells of the isthmus is (green, upper panel) and nuclei counterstained with DAPI (blue, lower panel). **c** and **e** scale bars represent 50 μm. Representative examples of three independent experiments are shown.

tube[10]. Finally, and similar to experiments in other mammals[7,22], tissue sections of wild type and *Adgrd1*$^{-/-}$ oviducts throughout the oestrus cycle failed to reveal a constriction or occlusion at the AIJ that could explain embryo retention, and this was consistent with the ease of flushing trapped embryos from the ampulla of *Adgrd1*$^{-/-}$ oviducts through the isthmus.

**Attenuation of post-ovulatory fluid flow is dysregulated in Adgrd1-deficient mouse oviducts.** Oviductal fluid influences embryo transport and varies throughout the oestrous cycle, peaking at ovulation and reducing in volume prior to the next cycle[23–27]. Due to the small size and flexuous morphology of the mouse oviduct, ligation is a practical method of evaluating fluid production[27], and so heterozygous *Adgrd1*$^{+/-}$ oviductal explants were ligated at the infundibulum in vitro at 0.5 dpc and 1.5 dpc. Four hours after ligation of 0.5 dpc oviducts, we observed a striking distention of the ampulla due to the accumulated fluid (Fig. 4a). This demonstrated that fluid exits the oviduct at the ovarial end creating a flow that opposes the movement of embryos - agreeing with observations in mice[27], and larger mammals[22,24,28,29].

Consistent with the reduction in fluid production after ovulation, ligating heterozygous oviducts in the post-ovulatory period (1.5 and 2.5 dpc) resulted in a much reduced accumulation of fluid (Fig. 4b and c). *Adgrd1*$^{-/-}$ mutant oviducts ligated at 0.5 dpc exhibited the same ampullary distention (Fig. 4a); however, by contrast to heterozygous littermates, did not show the same reduction at 1.5 and 2.5 dpc demonstrating that the post-ovulatory attenuation of fluid production in mutant oviducts is dysregulated (Fig. 4b and c). This was confirmed by performing the ligations in vivo at 2.5 dpc, which resulted in a more conspicuous distension of the ampulla due to the accumulated fluid in the mutant compared to heterozygous controls (Fig. 4d and Supplementary Fig. 3a). Dysregulated fluid production was specifically observed in the isthmus by ligating oviducts within the isthmus itself (Supplementary Fig. 3b). While these ligation experiments demonstrated defects in oviductal fluid production, they did not directly show how oviductal fluid flow was altered in Adgrd1-mutant oviducts. To address this, we developed a surgery setup similar to that used by Hino and Yanagimachi[27] using a heating jacket that permitted experimental access to the female reproductive organs to inject a tracer dye into the oviduct lumen and directly observe fluid

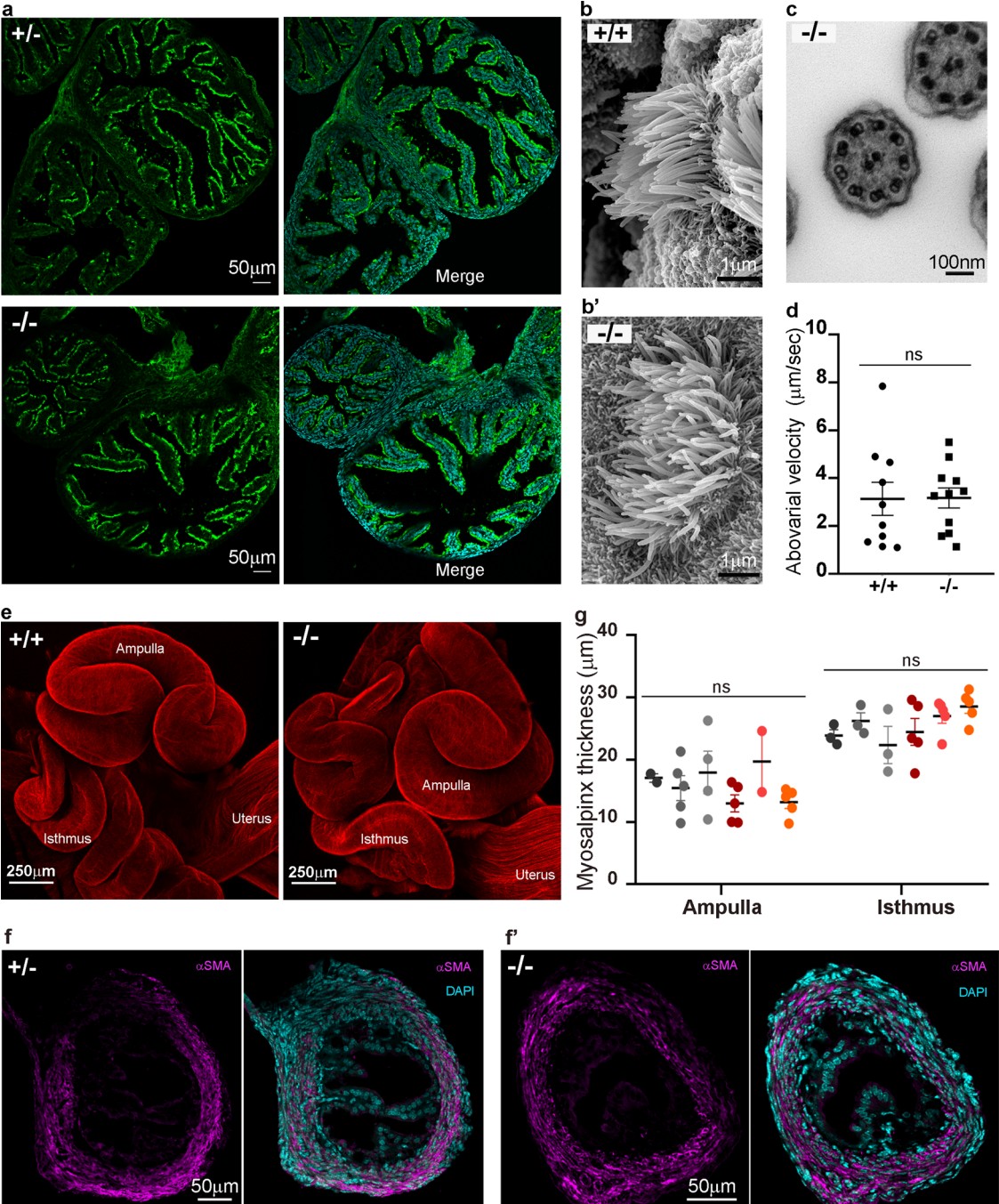

**Fig. 3 *Adgrd1*-deficient oviducts have morphologically normal cilia and muscle. a** The distribution of ciliated cells is similar in the epithelium of control *Adgrd*$^{+/-}$ and *Adgrd1*$^{-/-}$ oviducts. Oviductal sections from adult females in diestrous were stained with an antibody against acetylated tubulin to mark cilia (green), and nuclei counterstained with DAPI (blue, right panel). **b**, **b'** Ciliated cells analysed by scanning electron microscopy did not differ in mutant *Adgrd1*$^{-/-}$ oviducts compared to wild-type controls. **c** Transmission electron microscopy images of *Adgrd1*$^{-/-}$ cilia showed the usual 9 + 2 organisation of microtubules. **d** Polystyrene beads placed on oviductal epithelium explants moved at equivalent speeds in an aboarial direction in both control and *Adgrd1*$^{-/-}$ epithelial tissues. Individual data points are bead velocity quantified with Image J manual tracking for a minimum of 15 s. Bars represent the mean ± SEM, measurements were performed on 3 *Adgrd*$^{+/+}$ and 3 *Adgrd1*$^{-/-}$ ampullae; an unpaired t-test analysis showed no significant (ns) difference between the groups. **e** Image shows 3D projection of oviducts from 17-day-old mice stained with a phalloidin-Texas Red conjugate, demonstrating no overt difference in muscle structure and organisation between wild type and mutant. **f** and **f'** Representative examples of sections of the isthmus stained with anti-smooth muscle alpha actin (magenta) and counterstained with DAPI (Cyan). **g** The thickness of the myosalpinx is similar in controls (hues of grey) and *Adgrd1*$^{-/-}$ (hues of red) in the different oviductal regions. Bars represent the mean ± SEM, measurements were performed on 3 *Adgrd*$^{+/-}$ and 3 *Adgrd1*$^{-/-}$ oviducts; a minimum of two sections per mouse were analysed. A two-way ANOVA analysis found that the genotype has no effect (ns) while the difference between the regions of the oviduct is extremely significant ($p < 0.0001$). **a**, **b**, **b'**, **c**, **e**, **f**, and **f'** are representative examples of three independent experiments.

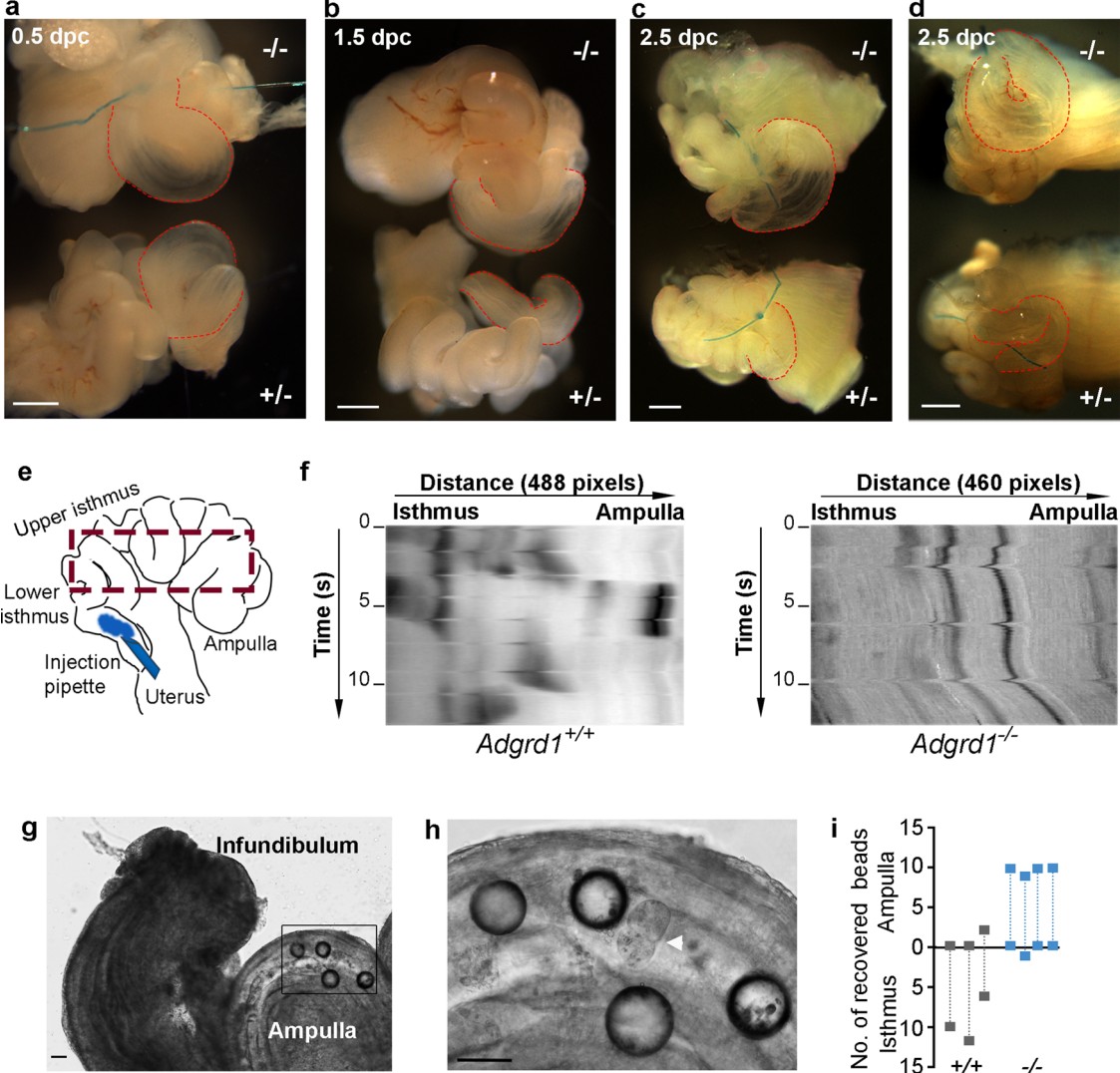

**Fig. 4 Adgrd1 regulates post-ovulatory attenuation of oviductal fluid production. a** Accumulation of fluid is comparable in ligated control (+/−) and mutant (−/−) oviductal explants in vitro at 0.5 dpc; ampullary distension is marked with red lines. **b** No reduction in fluid accumulation in mutant *Adgrd1*−/− oviducts at 1.5 dpc by contrast to heterozygous control. **c** Oviductal ligation at 2.5 dpc in vitro: more fluid has accumulated in the ampulla of the mutant compared to control. **d** The ampulla of mutant (−/−) oviducts exhibited a more pronounced distension compared to heterozygous control (+/−) after in vivo ligated at 2.5 dpc and collected four hours later; the ampullary region is highlighted by dotted red lines. Images in **a**, **b**, **c** and **d** are representative of at least three independent experiments and the scale bar represents 500 μm. **e** Schematic showing the region of the oviduct used for producing the kymographs. **f** Kymographs showing the behaviour of a tracer dye injected into the oviduct of Adgrd1-mutants compared to wild type littermate control. Adgrd1-mutant oviducts are rapidly filled with the dye which does not change over the time of observation whereas control oviducts segregate the dye into boluses which are gradually moved towards the ampulla. **g** Glass beads (~100 μm diameter) were inappropriately retained in the ampulla of *Adgrd1*−/− oviducts. Image taken 24 h after bead transplantation. **h** Magnification of the boxed area in **g**, showing glass beads and retained embryo (arrowhead). Scale bars represent 80 μm, images are representative of four independent experiments. **i** Distribution of 10 to 12 transplanted glass beads in *Adgrd1*−/− (n = 4) and control (n = 3) oviducts at 1.5 dpc.

movement in situ while maintaining a physiological temperature, blood circulation and tissue innervation (Supplementary Fig. 4). We focussed on comparing the fluid flow in Adgrd1-deficient and control oviducts at 1.5 dpc where the earlier ligation experiments had demonstrated differences. Consistent with findings from other groups and our ligation experiments, we observed adovarial flow of oviductal fluid in both Adgrd1-deficient and control oviducts (Supplementary Fig. 5 and Supplementary Movie 3). In the control animals, the dye behaved similarly to that observed by Hino and Yanagimachi: the dye segregated into several boluses within the oviduct close to the uterus which then gradually moved towards the ovary in a saltatory manner that preserved the quantised nature of the dye distribution (Supplementary Movie 3). By contrast, in

the Adgrd1-deficient oviducts, the tracer dye instantaneously dispersed along the entire length of the oviduct so that once filled, there was relatively little change in the distribution of the dye along the length of the oviduct over the course of the observation (Supplementary Fig. 5 and Supplementary Movie 3). To visualise these different behaviours, we drew kymographs which emphasised the pulsatile character observed in the fertile controls compared to the continuous rapid flow in the Adgrd1-deficient oviducts (Fig. 4e and f). Given these differences in fluid flow, we next asked if there were differences in secretory cell development in Adgrd1-mutant oviducts. Using a marker of secretory cells, we observed no difference in either the relative number or organisation of these cells within the mutant epithelium (Supplementary Fig. 6). Finally, we

reasoned that if embryo retention was due to dysregulated oviductal fluid production, the transport of particles other than embryos would be affected. Consistent with this, we found that appropriately-sized glass beads were similarly retained within the ampulla of $Adgrd1^{-/-}$ but not control oviducts (Fig. 4g, h and i). Together, these results show that the postovulatory cessation of oviductal fluid flow is misregulated in the oviduct of Adgrd1-deficient mice which could prevent embryo passage of the AIJ and cause infertility.

**The ADGRD1 ligand PLXDC2 is expressed on cumulus cells.** Adgrd1 encodes a cell surface receptor belonging to the adhesion G-protein coupled receptor family, a subset of over 30 proteins that typically contain a large N-terminal ectodomain, most of which have no identified ligand[30]. Adgrd1 contains a pentraxin domain in its extracellular region and is known to initiate intracellular signalling by stimulatory G proteins leading to increases in cAMP levels by activating adenylate cyclase[19]. One mechanism to trigger G-protein signalling used by this family of receptors is through the ligand-dependent relief of an auto-inhibitory ectodomain[30-32]. To identify a ligand for ADGRD1, we first expressed the entire ectodomain as a multimeric binding probe to increase binding avidity and thereby circumvent the often weak binding affinities of extracellular receptor-ligand interactions[33]. The highly avid ADGRD1 binding probe was then systematically tested in an unbiased manner for binding to a panel of 1132 unique human receptor ectodomains[34], and the Plexin Domain-Containing Protein 2 (PLXDC2) was identified as a candidate ligand (Supplementary Fig. 7a). To investigate this further, we used an assay designed to detect direct extracellular receptor-ligand interactions called AVEXIS[35] and demonstrated that ADGRD1 and PLXDC2 interacted in both bait-prey orientations (Fig. 5a). To further validate the interaction, we expressed the entire ectodomains of the mouse orthologues of both Adgrd1 and Plxdc2, and showed that they could directly interact (Fig. 5a). Cells transfected with a plasmid encoding Plxdc2 gained the ability to bind the highly avid Adgrd1 binding probe, and this binding was specifically blocked by preincubating the transfected cells with an anti-Plxdc2 antibody (Fig. 5b). We next mapped the interaction interface to the pentraxin domain of Adgrd1 and the PSI domain of Plxdc2 by creating a series of truncated ectodomains that encompassed known domains of both proteins and using the AVEXIS assay to quantify binding (Fig. 5c, d and Supplementary Fig. 7b). To understand how the Adgrd1 receptor could be activated in the oviduct, we characterised the tissue expression patterns of Plxdc2. Within the female reproductive system, *Plxdc2* was most highly transcribed in the ovary and cumulus-oocyte complexes (COCs) compared to the uterus and oviduct and other tissues such as brain, liver and spleen (Supplementary Fig. 7c). We confirmed the expression of Plxdc2 protein in the ovary and COCs by Western blotting (Fig. 5e), and showed using immunocytochemistry that Plxdc2 was highly expressed on the surface of cumulus cells (Fig. 5f). We demonstrated that Plxdc2 is an activating ligand for Adgrd1 using an established GPCR activation assay which measures the increase in cellular cAMP levels[36]. Adgrd1-transfected HEK293 cells showed an expected increase in cAMP levels[37] which was augmented by Plxdc2 (Fig. 5g). Together, these results identify Plxdc2, which is expressed on cumulus cells as an activating ligand for Adgrd1.

## Discussion

Ectopic pregnancy is a relatively common complication which affects up to 2% of all pregnancies in the United States and Europe and is a major risk factor for maternal health. Despite being so prevalent, we know remarkably little about the underlying causes which has prevented progress on developing treatments and diagnostics. The currently known risk factors are generic (previous IVF treatment, smoking, damage to the Fallopian tube due to surgery or infection) and therefore lack any insight into the underlying mechanisms involved. The few genetic studies in humans have reported dysregulated expression of genes at the site of ectopic implantation including HOXA10[38], leukaemia inhibitory factor[39], and MUC1[40], but these observational studies do not distinguish between cause and consequence. Female mice lacking functional cannabinoid receptor1 (*Cnr1*) have reported defects in embryo transport, although the phenotype differs from Adgrd1 mutant mice because they are only subfertile, with 65% of mothers still showing embryo implantation within the uterus. Moreover, the defect results in a delay of embryo transport rather than complete block, because embryos are recovered from the isthmus and are not retained within the ampulla[41]. Our finding that female mice lacking functional Adgrd1 are sterile due to embryo retention in the ampulla provides strong evidence that embryo transport is genetically controlled.

Beyond unequivocally demonstrating a genetic basis for embryo transport, our results also provide a mechanism to explain the valve-like behaviour of the mammalian oviduct. We present a model (Fig. 6) where ovulated COCs are propelled towards the uterus by ciliary action but are halted at the AIJ due to the opposing force of oviductal fluid flowing towards the ovary caused by the narrowing of the oviduct at the isthmus. Adgrd1 is locally activated in the oviductal epithelium by Plxdc2 displayed on the surface of cumulus cells that are progressively released as the jelly-like hyaluronic acid matrix surrounding the cumulus mass gradually disintegrates. The triggering of Adgrd1 decreases oviductal fluid production and consequently the flow that opposes the constitutive aborvial ciliary action to permit continued passage of embryos through the isthmus. This suggests that cumulus cells not only support oocyte development, but also communicate with the oviduct to regulate tubal transit. While the flow of the oviductal fluid towards the ovary which opposes the movement of embryos appears counterintuitive, this behaviour has been reported in several mammals and likely assists the ascent of sperm from the storage regions in the isthmus towards the site of fertilisation. It was first reported in larger animals such as cattle[22], sheep[24] and rabbits[28] where the surgical implantation of cannulas can be achieved more easily, but more recently observed directly using video microscopy in mice[27]. In mice, the fluid empties out into the peritoneal cavity through a hole in the ovarian bursa and the rate of flow is surprisingly high, estimated at 2.2 microlitres per hour in the periovulatory period but decreasing by 1.5 dpc[27].

One area for future investigation is to understand how Plxdc2 mediated activation of Adgrd1 and the consequent increase in cAMP within the oviductal epithelial cells leads to changes in fluid flow to regulate embryo passage. Consistent with our findings, prior research using perfused and cannulated oviducts has shown that increasing cAMP concentrations by adding a cell permeable analogue of cAMP (dibutyryl cAMP) or agents that increase cAMP levels such as forskolin, theophylline and cholera toxin all resulted in a decrease or abolition of oviductal secretion in humans[42] and rabbits[23,43]. These results are in keeping with a regulatory role for fluid secretion and the fact that we observed no changes in the relative number and organization of the secretory cells in the oviducts of Adgrd1-deficient mice. The addition of inhibitors of ion channels that may be regulated by cAMP levels to oviducts such as bumetanide and 5-nitro-2- (3-phenylpropyl-amino) benzoic acid did not provide any further insight, but these experiments were compromised because they were performed on explanted oviducts which are separated from the circulation and innervation. One possibility is that fluid flow could be regulated

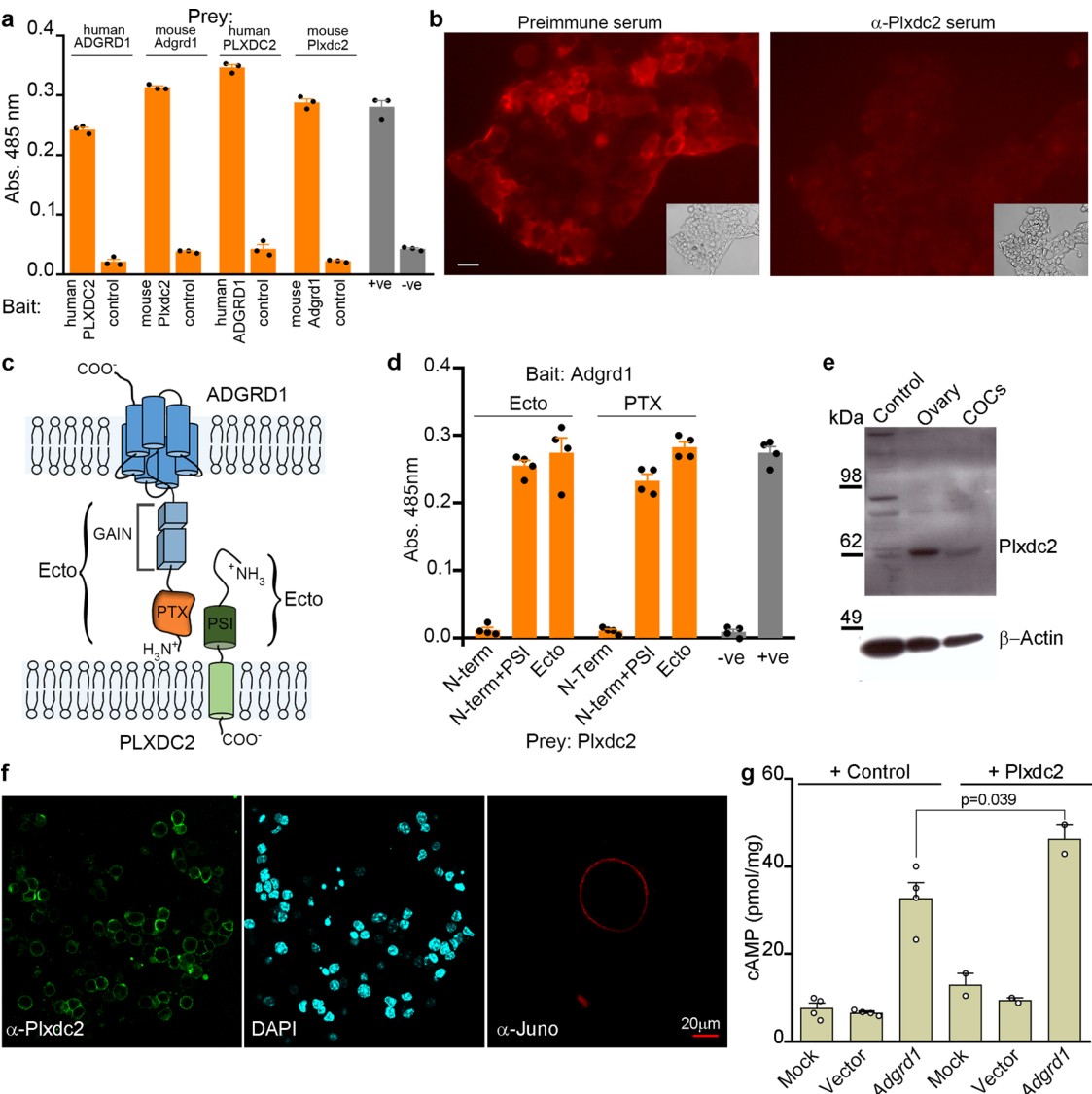

**Fig. 5 Plxdc2 is an activating ligand for Adgrd1 and expressed on cumulus cells. a** Direct interactions between human and mouse ADGRD1 and PLXDC2 ectodomains in both bait-prey orientations using the AVEXIS assay ($n = 3$ independent replicates). **b** Adgrd1 ectodomain probe bound *Plxdc2*-transfected cells (left panel) and was blocked by preincubation with Plxdc2 antiserum (right). Scale bar represents 10 µm. **c** Schematic of ADGRD1 and PLXDC2 domain organisation. **d** Domain truncations demonstrate that the pentraxin (PTX) domain of Adgrd1 and PSI domain of Plxdc2 are sufficient for binding ($n = 4$ independent replicates). **e** Western blotting and immunofluorescence **f** demonstrates that Plxdc2 is expressed on the plasma membrane of cumulus cells. HEK293T cells were used as negative control in **e**, anti-Juno shows the localisation of the oocyte in **f**. **g** The overexpression of *Adgrd1* induces a significant increase of intracellular cAMP compared to non-transfected cells (mock) and to cells transfected with a plasmid encoding GFP (vector). The Plxdc2 ectodomain induces a significant increase of cAMP levels in *Adgrd1*-expressing cells only. ($n = 2$ independent experiments). Bars in **a**, **d**, and **g** represent mean ± SEM.

by changes in the tone of the myosalpinx to control the rigidity of the oviduct. In our characterisation of the Adgrd1-deficient oviducts, however, we observed no morphological or developmental defects within the oviductal smooth muscle, and when observed in situ, the contractions appeared completely normal in terms of their frequency, strength, and localisation. It has been convincingly shown by others that ablating oviductal muscle contractions using nicardipine had no effect on the rate of oviductal fluid flow[27] suggesting no direct role of muscle in regulating oviductal fluid flow. Similarly, previous attempts to explain the tubal-locking phenomenon led to careful histological examination of the muscles at the AIJ, and did not reveal a sphincter-like organization[7,22], and complete resection of the AIJ did not affect the fertility of rabbits[44]. If the myosalpinx does play a regulatory role in fluid flow, then these changes are likely to be subtle and

gradual, and will therefore require advanced longer term in vivo imaging to investigate.

The roles of PLXDC2 and ADGRD1 in humans are poorly characterised and our findings warrant further investigations, particularly for any role they may have in reproduction and ectopic pregnancy. Both genes are additionally expressed outside of the female reproductive system suggesting other functions, and systematic mouse knockout phenotyping have reported bone mineralisation phenotypes as well as female sterility for *Adgrd1* (see the significant phenotypes reported by the International Mouse Consortium https://www.mousephenotype.org/data/genes/MGI:3041203). Mice containing a transgene targeting the majority of the signal peptide of *Plxdc2* resulted in a ten-fold decrease in *Plxdc2* transcript levels in the cerebella[45] but no abnormal phenotypes were reported. One suggested possibility is

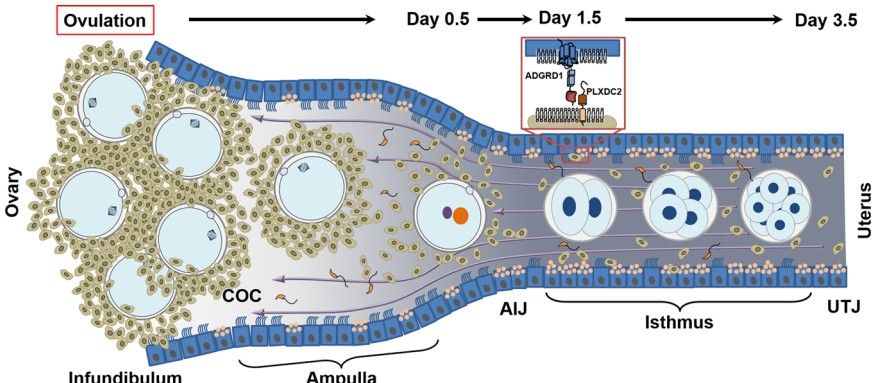

**Fig. 6 Model describing Adgrd1-mediated control of fluid flow underlies the valve-like behaviour of the mammalian oviduct.** Cumulus-oocyte-complexes are ovulated into the infundibulum and rapidly transported to the AIJ by the action of the cilia lining the oviductal epithelium beating towards the uterus. At the AIJ, the COCs are halted due to the balance of aboarial ciliary action and the force of adovarial oviductal fluid flow (purple arrows). In wild-type oviducts, the gradual release of cumulus cells as the hyaluronic acid matrix which surrounds them slowly disintegrates releases the Plxdc2 ligand and triggers a reduction of oviductal fluid production by activating Adgrd1 on the oviductal epithelium to licence embryo release. COC = cumulus-oocyte-complex; AIJ = ampullary-isthmic junction; UTJ = uterotubal junction.

that a gene encoding a related protein, Plxdc1, might compensate for the role of Plxdc2. Genome-wide association studies have implicated polymorphisms linked to *ADGRD1* with variations in adult height[46,47] and heart beating frequency in electro-cardiograms[48]. Subsequent studies focussed on *ADGRD1*, reported no clinical phenotypes in adults that are heterozygous for predicted loss of function alleles consistent with our findings in mice[49]. Importantly, tubal ectopic pregnancies are restricted to primates and are not observed in other animals[1]. In mice, embryos recovered from the ampulla of Adgrd1-deficient mice had not developed beyond the blastocyst stage in agreement with the reported ability of the ampulla to sustain the development in vitro to that stage[50], and suggesting that the ampulla cannot support embryo implantation and further development.

Our demonstration that Adgrd1 is essential for embryo transport provides strong evidence that embryo transport is genetically controlled, and provides a plausible mechanism to explain how the tubal locking phenomenon described many years ago in the mammalian oviduct is regulated. Ectopic pregnancy is a common and potentially deadly condition for which we have very little mechanistic understanding, and so our findings here not only provide potential starting points to develop diagnostics and treatments but also an animal model to study this condition further.

## Methods
**Generation, breeding and fertility phenotyping of *Adgrd1*-deficient mice.** All animal experiments were performed under UK Home Office governmental regulations and European directive 2010/63/EU. Research was approved by the Sanger Institute Animal Welfare and Ethical Review Board. *Adgrd1*$^{-/-}$ mice were obtained from the Knockout Mouse Project resource (IKMC Project: 22527) and contain a lacZ-tagged allele targeted to the *Adgrd1* genomic locus located on chromosome 5, *Adgrd1*$^{tm1b(EUCOMM)Wtsi}$. Mice were generated by injecting blastocysts with the targeted mouse embryonic stem cells which were transferred to pseudopregnant females to generate chimeras. Germline transmission of the targeted allele was confirmed by PCR after mating of chimeric males with C57BL/6NTac females. To obtain the reporter-tagged deletion allele, females heterozygous for the 'knockout-first' allele *Adgrd1*$^{tm1a(EUCOMM)Wtsi}$ were crossed to hemizygous males ubiquitously expressing the Cre enzyme[51]. Mice with the recombined allele were identified using PCR and diagnostic primers using genomic DNA extracted from ear biopsies (4403319 DNA Extract All Reagents Kit, Life Technologies) as the template for short range PCR using Platinum® Taq DNA Polymerase (10966034 Invitrogen). The mouse colony was maintained by crossing heterozygous males and females. Male and female fertility was quantified by pairing homozygous and heterozygous *Adgrd1* transgenic adult mice with homozygous and heterozygous animals of proven fertility and the number of resulting pups was monitored continuously for three months. The number of ovulated oocytes was counted after induction of ovulation with 5 IU of pregnant mare serum

gonadotropin (PMSG) followed by 5 IU of human chorionic gonadotropin (hCG) 48 h later. In vivo fertilisation was assessed by scoring the number of zygotes and the number of non-fertilised oocytes present in the ampulla at 0.5 dpc. Eggs and embryos were fixed in 4% formalin (28906 Thermo Scientific Pierce) and stained with DAPI (62248 Thermo Fisher). In vitro fertilisation was performed essentially as described[52]. Briefly, sperm were collected from the cauda epididymis of adult male mice, capacitated for 1 h in HTF medium at 37 °C and added to cumulus-enclosed oocytes that had been collected 13 h after hCG treatment. Groups of 20 to 30 eggs were inseminated in 100 μL drops of HTF medium containing $1 \times 10^5$ sperm; four hours later, eggs were washed and cultured in KSOM (MR-121D EmbryoMax, Millipore) for the following four days. Formation of pronuclei was scored and embryo development recorded daily. Mutant and age-matched control females were mated with proven males and the day of vaginal plug formation was counted as 0.5 dpc. The female reproductive tracts were visually inspected for the presence of implantation sites at 5.5 dpc, 6.5 dpc and 8.5 dpc. The embryo position in the oviducts was determined using the Zeiss Axioplan 2 microscope and embryos were counted after flushing the oviducts with M2 medium (Sigma).

Non-surgical embryo transfer (NSET) was performed as described[53]. Wild-type 2-cell embryos were cultured in KSOM for 48 h and blastocysts were transferred to pseudopregnant *Adgrd1*$^{+/+}$ and *Adgrd1*$^{-/-}$ recipient females at 2.5 dpc using the NSET device (Paratechs, catalogue #60010). 14 and 16 blastocysts were transferred to two wild-type females, and 14, 17, 29 and 30 blastocysts were transferred to four *Adgrd1*-deficient females. The number of implanted embryos was counted ten days after the transfer.

**Time-lapse video microscopy.** Oviducts were carefully dissected from 8-week-old females at the required stage of the oestrus cycle. Entire oviducts were separated from the ovary by opening the ovarian bursa, and scission of the ovarian ligament; isolation from the rest of the reproductive tract was achieved by trimming the broad ligament whilst maintaining the utero-tubal junction. When cutting the mesosalpinx, care was taken to avoid pulling and damaging the tubes. Spontaneous rhythmic contractions of oviducts were documented by video recording immediately after dissection using a Zeiss Axioplan 2 microscope equipped with a Hamamatsu 1394 ORCA-ERA digital camera and the Velocity imaging software. Fifty seconds of real time were acquired at a frame frequency of 1 Hz. To record ciliary beating, the ampullary region of the oviducts were opened longitudinally, placed in PBS on a microscope glass and microparticles (Sigma 74964) were gently added to the tissue. Movies were acquired at 8.5 frames per second.

Micro beads (Micro particles based on polystyrene, 15μm of diameter; no. 74964 Merck Life Science) were applied to longitudinally opened ampullae in DPBS with Ca$^{2+}$ and Mg$^{2+}$, and live videos were recorded with a stereomicroscope (Leica M205FA) equipped with the Leica DFC7000T camera. The velocity of bead transport was quantified with Image J manual tracking for a minimum of 15 s per bead.

**Visualisation and analysis of the oviductal fluid flow.** The design of the plastic jacket was drawn with AutoCAD Inventor and printed with Ultimaker 2+ Connect 3Dprinter using a PolySmooth printing filament. After printing the surface was smoothed with Isopropanol and the lid was sealed, finally the surface was treated with the Polymaker Polysher from 3DJake UK to prevent leakage. In preparation for surgery the jacket was connected to a motor water pump immersed in warm water. For the visualisation of oviductal fluid flow female mice were mated, selected at the required time, and prepared for surgery as described above. The ovary and

the oviduct were exposed on the surface of the surgical incision and manipulated with care to avoid damaging the tissues. A topical tissue adhesive (Gluture, Zoetis) was used to hold the fat pad and the ovary onto the skin on one side of the incision. The jacket containing circulating warm water was sealed to the mouse skin and pre-warmed DPBS with $Ca^{2+}$ and $Mg^{2+}$ (Hyclone SH30264) was added to maintain the tissues at ~37 °C. A 40 μm bevelled glass pipette (BioMedical Instruments VESbv-40-0-0-55) was used to inject a tracer dye (India Ink) in the lower isthmus. The dye was dialysed overnight in DPBS and diluted twofold in the same buffer before injections. Videos were recorded with the Leica MC170HD camera at 24fps and retrospectively analysed. Kymographs were generated with the dedicated Image J plug-in by selecting a region encompassing the oviducts for a total of 12.5 seconds.

**Wholemount LacZ staining, immunofluorescence and immunohistochemistry**. To determine the expression of the lacZ reporter gene in the targeted allele at the *Adgrd1* locus, dissected tissues were fixed in 10% Neutral Buffered Formalin (NBF, Cellpath) overnight, washed with PBS, stained with a LacZ staining solution (2 mM $MgCl_2$, 0.02% IGEPAL CA-630, 5 mM potassium ferrocyanide, 5 mM potassium ferricyanide, 0.01% deoxycholic acid, 0.1% X-Gal (5-bromo-4-chloro-3-indolyl β-D-galactopyranoside) in dimethylformamide) at 4 °C, post-fixed in 4% formalin and stored in 70% glycerol.

Polyclonal antiserum was raised by immunising rabbits with the entire ectodomains of mouse ADGRD1 or mouse PLXDC2 expressed as soluble recombinant 6-his-tagged proteins in HEK293-6E cells and purified using $Ni^{2+}$-NTA chromatography, essentially as described[54]. Immunisations were carried out by Cambridge Research Biochemicals in accordance with UK Home Office regulations and the Sanger Institute animal welfare ethical review board.

To prepare wholemount tissues for immunofluorescence, they were fixed in 4% formalin for 30 min at room temperature, incubated with the primary antibody for 1 h, and washed extensively with PBS and finally stained with the secondary antibody conjugated with the appropriate fluorochrome (Alexa Fluor 488 goat anti-mouse supplied by Thermo Fisher, and Alexa Fluor 647 goat anti-rabbit supplied by Jackson Immunoresearch). Where tissues were sectioned, tissue was fixed in 4% formalin for 60 min, rinsed extensively in PBS and soaked in a sucrose solution overnight before embedding in OCT and 8 micron sections cut with a Leica CM1950 cryostat. Sections were treated with 1% SDS in PBS for 5 min, blocked before adding primary antibody overnight at 4 °C. Sections were washed and incubated with a goat anti-rabbit – Alexa488 secondary antibody for 1 h at room temperature. Sections were then washed and mounted in slowFade Gold mounting solution with DAPI (Thermo).

Binding of recombinant proteins to cells was quantified by incubating transfected cells with highly avid ectodomains expressed as FLAG-tagged pentameric preys in cell culture medium at 37 °C before fixation in a phosphate-buffered 4% formalin solution. Staining of muscle was performed by first permeabilizing fixed oviducts with 0.2% Triton X-100 and incubating with Texas-Red-phalloidin conjugate for 30 minutes. Optical sections of whole oviductal tissue were acquired with a Leica SP5 laser confocal microscope (z-step size = 2.52μm). The 3D projection of the whole z-stack was generated with the LAS AF software. The antibodies and probes and the dilutions used in this study were: polyclonal rabbit-anti-mouse Adgrd1 (this study) 1:100; polyclonal rabbit-anti-mouse Plxdc2 (this study) 1:100; rat monoclonal anti-mouse Juno (clone TH6, Biolegend) 1:200; mouse monoclonal anti-acetylated Tubulin (clone 6-11B-1, Sigma Aldrich); anti-Pax8 rabbit polyclonal antibody (10336-1-AP, Proteintech) 1:400; mouse monoclonal anti-alpha smooth muscle Actin (clone 1A4, A2547, Sigma) 1:400; mouse monoclonal anti-Flag-Cy3 conjugate (clone M2, Sigma Aldrich) 1:500; Texas-red conjugated phalloidin (Thermo Fisher, T7471); Alexa Fluor 633 goat anti-mouse IgG (Thermo Fisher A-21052)1:500; Alexa Fluor 488 goat anti-mouse IgG (405319 Biolegend) 1:500; Alexa Fluor 488 goat anti-rabbit IgG (111-545-003 Jackson Immunoresearch) 1:500; pentameric FLAG-tagged PLXDC2 (this study) 1:10.

**Transmission and scanning electron microscopy**. Mouse oviducts were fixed at room temperature in a 2% formalin/2.5% glutaraldehyde mixture buffered in 0.1 M sodium cacodylate at pH 7.4 for one hour, rinsed and fixed in 1% osmium tetroxide for another hour followed by 1% tannic acid and 1% sodium sulphite for 30 minutes respectively. Samples were then dehydrated in an ethanol series, staining en bloc with uranyl acetate at the 30% stage before embedding (Epoxy embedding medium kit – Sigma). 50 nm ultrathin sections were cut on a Leica UC6 ultra-microtome and imaged on an FEI 120 kV Spirit Biotwin TEM with a F4.15 Tietz camera.

For scanning electron microscopy, mouse oviducts were fixed as for TEM (replacing tannic acid with osmium-thiocarbohydrazide), dehydrated in an ethanol series, critical point dried in a Leica CPD300, mounted and sputter-coated with 2 nm of platinum using a Leica ACE600 evaporator. Samples were imaged on a Hitachi SU8030 SEM.

**Oviduct ligation, surgical procedures and transfer of beads**. Oviducts from mice at the required stage in the oestrus cycle were dissected under a Nikon SMZ800 stereo microscope. Once the ovary was separated from the oviduct and

removed, a surgeon's knot was tied around the oviduct with polypropylene suture (Ethicon Prolene 10-0, W2794). The ligated oviducts were maintained in culture at 37 °C in 500 μL of pre-warmed KSOM media, fixed in NBF overnight, and imaged with a Leica M205FA stereomicroscope.

Glass beads with a diameter of ~100 μm (Sigma G4649) for transfer into oviducts were sterilised by autoclaving and washed in M2 medium. Female recipient mice whose mating had been timed were prepared for surgery using gaseous anaesthesia under aseptic conditions, shaved, and given analgesia. Using a stereomicroscope (Leica MZ7.5), a small longitudinal skin incision was made at the midline, level with the last rib. The ovary was located through the muscle wall and micro scissors were used to make a small incision, followed by blunt dissection and exteriorisation of the ovary and oviduct by holding the associated fat pad, anchored using a serrefine clamp. After removing the bursa covering the ovary and oviduct, glass beads were taken up using a mouth pipette into a 230 mm Pasteur glass pipette with the aid of a stereomicroscope (Leica MZ9.5) and transferred into the oviducts of recipient mice at 0.5 dpc. The location of the beads in the ampulla and isthmus were counted at 1.5 dpc. In vivo ligation of oviducts was performed using the same surgical procedure with polypropylene suture which was slid under the oviduct ~1 mm from the infundibulum and fine forceps used to tie a surgeon's knot. When assessing fluid accumulation in the isthmus, additional ligatures were tied at the ampullary-isthmic junction and the isthmus. The ovary and oviduct were carefully replaced within the body cavity using blunt forceps, and the skin incision closed using a single wound clip.

**Preparation of the protein library for large-scale receptor screening**. A library of single-transmembrane-spanning (STM) human proteins was compiled using computational prediction algorithms and experimental evidence, as described[25]. The ectodomain of each protein was synthesised and cloned into the pRK5 vector (Genentech) in frame with a C-terminal Fc (hIgG1) tag. The resulting library consists of 1364 human proteins, including 1132 unique receptors and a number of replicate constructs as assay controls. Expression of the human STM protein library was performed as described[34]. In brief, conditioned media enriched in individual receptors was prepared in Expi293F cells (Thermo Fisher), transiently transfected with receptors expressed as ectodomains fused to a human Fc (IgG1). Expi293F cells were cultured in Expi293 Expression Media (Cell Technologies) in flasks at 37 °C and 150 r.p.m. agitation in a humidified incubator. Human cells were chosen for expression of the collection of STM receptors to maximise protein quality and incorporation of relevant post-translational modifications. One mL cell transfections were performed using an automated system consisting of a TECAN liquid handling system and a MultiDrop reagent dispenser. Transfections were processed in batches of 96 clones by a Biomek FX liquid handling robot and conditioned media was harvested 7 days post-transfection. Subsequently, the receptor Fc-tagged ectodomains present in the conditioned media were captured on protein A-coated 384-plates (Thermo Scientific), and stored at 4 °C until use. For the binding screen, the ectodomain of ADGRD1, truncated at S600, was expressed in HEK293-6E cells as a pentameric probe fused to a beta-lactamase enzyme[55]. A Thr to Gly mutation at position 574 in the GPCR proteolytic site (GPS) of human ADGRD1 was introduced to prevent the autoproteolysis[56].

**Identification of binding extracellular ligands for ADGRD1**. The Cell Surface Receptor Interaction screening was based on the AVEXIS method[35], which was further implemented for automated high throughput screening in 384 well plate format, as recently described[34]. In brief, screens were performed using an integrated robotic system consisting of automated liquid handling devices for high throughput analysis of receptor-ligand interactions. On the day of the assays, the protein A plates coated with the receptor library were washed with PBS containing $Ca^{2+}$ and $Mg^{2+}$, followed by incubation with the beta-lactamase-tagged penta-meric ADGRD1 probe for 1 h at room temperature. Plates were then washed to remove free protein prior to addition of nitrocefin (Calbiochem) for 1 h at room temperature. Hydrolysis of the beta-lactamase substrate nitrocefin was quantified by measuring absorbance at 485 nm using an integrated TECAN plate reader. The raw enzymatic absorbance values were analysed to calculate Z-scores across all proteins in the library. Comparisons to previous screens of this type identified immobilised ligands which repeatedly gave positive binding signals irrespective of the binding probes used, and included known lectins (e.g., SIGLEC family members and MRC1) which are likely to directly interact with common glycans presented on the binding probe.

**Recombinant protein production and protein interaction screening**. The extracellular domains of ADGRD1 and PLXDC2 were expressed as soluble secreted proteins by transient transfection in HEK293-6E cells[55]. NCBI reference sequences for the expressed proteins were: mouse Adgrd1: NP_001074811; human ADGRD1: NP_001317426; mouse Plxdc2: NP_080438; human PLXDC2: NP_116201. The region encoding the entire predicted ectodomain was chemically synthesised by gene synthesis (GeneArt, Life Technologies) flanked by unique NotI, AscI restriction sites to facilitate cloning into mammalian expression plasmids to produce enzymatically biotinylatable monomeric 'baits' or pentameric FLAG-tagged 'preys'; both bait and preys contained a rat Cd4d3 + 4 epitope tag as described. Biotinylated proteins were produced by co-transfecting a plasmid encoding a

secreted version of the BirA enzyme[55]. To produce recombinant proteins corresponding to specific regions of both ADGRD1 and PLXDC2 ectodomains, primers were designed that would amplify the appropriate region from the plasmid encoding the entire ectodomain by PCR and the products cloned into both bait and prey expression plasmids using unique NotI and AscI sites[54]. Bait proteins were normalised by ELISA using a monoclonal antibody (Ox68) recognising the rat Cd4d3 + 4 tag. Prey proteins were normalised to their β-lactamase activity to levels suitable for the AVEXIS assay as described[57]. Biotinylated baits that had been either purified or dialysed against PBS to remove excess free D-biotin were immobilised in streptavidin-coated 96-well microtitre plates (NUNC). Preys were incubated for one hour, washed three times in PBS/0.1% Tween-20 and once in PBS. Finally, 60 μl of 125 μg/ml nitrocefin was added and the absorbance measured at 485 nm on a Pherastar plus (BMG laboratories). A biotinylated protein consisting of just the rat Cd4d3 + 4 tag alone was used as a negative control bait and a biotinylated Ox68 monoclonal antibody (anti-prey) diluted 1:2,000 was used as an internal control. The rat Cd200-Cd200R interaction was used as a positive control.

**Western blot**. To perform Western blotting, proteins from mouse ovary, ovulated COCs, and HEK293-T were extracted with RIPA buffer and quantified with Bradford assay (Thermo) following manufacturer's instructions. Briefly, normalised protein amounts were resolved under reducing conditions by SDS–PAGE and blotted to Hybond-P PVDF membrane (GE Healthcare) for 1 h at 30 V. After blocking for 1 h with 2% BSA, the membrane was incubated for 1 h with Plxdc2 antiserum diluted 1:100 or with anti beta-Actin (Abcam, ab8227) diluted 1:1000 in PBST(PBS/0.1% Tween-20) added with 2% BSA, washed three times and incubated with a horseradish peroxidase (HRP)-conjugated anti-rabbit antibody (Thermo-Fisher Scientific Cat. No. G21234) diluted 1:5,000. Proteins were detected using SuperSignalWest Pico Chemiluminescent substrate (Thermo Scientific) and developed on photographic film. Uncropped blots can be found in the source data file.

**RT-PCR**. Total RNA was extracted from mouse tissues using Trizol reagent (Invitrogen) as per manufacturer's instructions, resuspended in water, and quantified with a Nanodrop 1000 spectrophotometer (Thermo Scientific). SuperScript III reverse transcriptase (Invitrogen) was used to produce cDNA from 1 μg of RNA and subsequent amplification was obtained with the KOD hot start DNA polymerase (Novagen). A list of all primers is provided in Supplementary Table 1. PCR products were resolved on a 1.5% agarose gel and imaged with an Azure c-600 gel documentation system.

**cAMP ELISA**. Adherent HEK293-T cells were transiently transfected with either a plasmid encoding mouse *Adgrd1*, or a negative control encoding membrane-tethered EGFP (vector), or were treated with the transfection reagent Lipofectamine 2000 alone (mock). Twenty-four hours after transfection, cells were seeded on a streptavidin-coated microtitre plate that had been pre-incubated with the mouse PLXDC2 or control rat Cd200 biotinylated ectodomains. After three hours at 37 °C, the levels of cyclic AMP were determined in duplicate using a cAMP ELISA kit (ADI-900-066, Enzo Life Science) according to the manufacturer's protocol, and normalised to the total protein concentration determined using the Bradford assay (ThermoFisher Scientific). One-way ANOVA showed a significant variation among conditions ($F_{5,12} = 43$, $p < 0.0001$) and a post-hoc Bonferroni test indicated that intracellular cAMP differed significantly in *Adgrd1*-transfected cells treated with Plxdc2 compared to *Adgrd1*-transfected cells treated with the control protein CD200 ($p = 0.0391$).

**Reporting summary**. Further information on research design is available in the Nature Research Reporting Summary linked to this article.

## Data availability

Data supporting the findings of this manuscript are available in the Source Data file and from the corresponding author upon reasonable request. Source data are provided with this paper.

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

## Acknowledgements

We thank the Sanger Mouse Genetics Programme for generating the *Adgrd1*⁻/⁻ mice and for the initial phenotyping; the staff at the Sanger Research Support Facility for the provision and help with the management of the mice; and Colin Barker from the Sanger Workshop for the design and building of the surgery jacket. This work was funded by the Medical Research Council (grant MR/M012468/1) and the Biotechnology and Biological Sciences Research Council (grant BB/T006390/1) to EB and GJW and Wellcome Trust (grant 206194) to GJW.

## Author contributions

E.B. performed all experiments unless stated. Mouse surgery was performed by M.W.; Receptor screening was performed by Y.S. and N.M.-M.; domain mapping experiments were performed by A.A.-O.; electron scanning and transmission microscopy was performed by DAG. The manuscript was written by E.B. and G.J.W.

## Competing interests

N.M.-M. owns shares in the Genentech/Roche group. The remaining authors declare no competing interests.
