## [Peer Review File · Nature Communications]

Reviewers' Comments:

Reviewer #1:

Remarks to the Author:

The reviewer thanks the authors to decipher how the cumulus cells regulate oviductal luminal fluid production and embryo transition through the ampullary-isthmic junction (AIJ). This cumulus mediated control of oviductal functions will be of interest to both basic and clinical researchers in the field. Although the manuscript is clearly written and technically sounds, this reviewer still has some comments to improve the manuscript.

1. Both *Adgrd1* and *Plxdc2* are expressed in several tissues. How can the authors exclude the possibility that the ADGRD1 (with PLXDC2) mediated signaling in the other tissues indirectly regulate the oviductal fluid production? Conditional KO or rescue by local gene delivery (eg., tissue-specific Tg, viral vectors, EP) would be best to answer. Alternatively, the authors may want to inhibit the pathway with antibody (or inhibitory domain) releasing beads, etc.
2. The authors demonstrated that the *Adgrd1* is expressed in the part of the oviduct but did not show ovulated COCs. The reviewer would like to know if *Adgrd1* is expressed in COCs or not. If yes, the authors should show if *Adgrd1* KO COCs are functional in oviductal fluid regulation. Although it is not an easy experiment, this can be addressed by transplanting the KO ovary into WT females. The reciprocal experiments are welcome.
3. There is no apparent abnormality reported in *Plxdc2* hypomorphic mutant mice (<http://www.informatics.jax.org/marker/MGI:1914698>). It should be addressed by cumulus cell-specific gene disruption (or complementation in whole KO), or at least discussed.
4. Does PLXDC2 function in membrane form or secreted form?
5. The readers may want to see the fallopian tube as extended in Figure 2a.

Reviewer #2:

Remarks to the Author:

This paper reports the findings of an investigation of the action of *Adgrd1* in relation to mammalian reproduction. The transport of gametes through the fallopian tube and fertilization in the fallopian tube is a highly regulated time sensitive process necessary for reproduction. Infertility, subfertility and ectopic pregnancy can result when tubal transport is adversely affected. This reports the novel findings of *Adgrd1* action by noting the response in deficient models, as well as localization within the oviduct. This is a impressive introductory work into this area. Future directions may include human fallopian tube specimens obtained from routine gynecologic surgeries (sterilization or hysterectomy). Very nice figure 4 to add to the discussion. Would like to see more in depth discussion of the clinical relevance in the discussion as opposed to a summary of the results, as well as discussion of limitations and future directions.

Reviewer #3:

Remarks to the Author:

The manuscript investigated mechanisms of how valve-like behaviors of ampullary-isthmic junction (AIJ) is controlled during the process of embryo transport in the oviduct. In the first part of the paper, it is clearly shown that the orphan adhesion G-protein coupled receptor *Adgrd1* is required for the passage of embryos into the isthmus from the ampulla where fertilization occurs. In *Adgrd1* mutant mice, embryos were arrested in the ampulla and reached the morula stage by E2.5. Transport of embryos through the oviduct is thought to be controlled by multiple factors including the unidirectional beating of cilia, contractions of muscles, and fluid secretion. The authors analyzed the effects of *Adgrd1* mutation on those factors and proposed that dysregulation of fluid secretion in the *Adgrd1* mutant oviduct caused the arrest of embryos in the ampulla. They finally explored molecules which activate *Adgrd1* in the oviduct and found a candidate, *Plxdc2*. Their finding that female mice lacking *Adgrd1* are sterile because embryos are arrested in the

ampulla is potentially interesting. However, the manuscript is less convincing, when they tried to validate the role of the fluid secretion on the passage of embryos through AIJ, which is the main point of the study. More direct evidence is needed to conclude that the control of oviductal fluid secretion by *Adgrd1* is essential for unlocking the restraining mechanism at the AIJ. Furthermore, the effects of *Adgrd1* mutation on cilia beating, muscle contraction, and tissue structures need to be more carefully and thoroughly analyzed. Those two major points require revisions.

Major points:

The most critical point is that the authors did not explain exactly why defects in fluid secretion are causative of the arrest of embryos in the *Adgrd1* mutant oviduct. More direct pieces of evidence are needed to explain the mechanisms of unlocking the valve-like "tubal-locking".

Changes in fluid secretion are at the base of the proposal of this paper. Yet, the data are not convincing enough; only a few examples are shown and the changes are not quantitatively and statistically assessed (Fig. 2 c-e and supplementary Fig.4).

Minor points:

Supplementary Figure 3a

Sufficient details of muscle structure and organization are not given.

Supplementary Movie 1 and 2

Differences in the cilia activity and the muscle contraction between the wild type and the mutant oviduct should be quantitatively and statistically assessed.

Reviewer #1

The reviewer thanks the authors to decipher how the cumulus cells regulate oviductal luminal fluid production and embryo transition through the ampullary-isthmic junction (AIJ). This cumulus mediated control of oviductal functions will be of interest to both basic and clinical researchers in the field. Although the manuscript is clearly written and technically sounds, this reviewer still has some comments to improve the manuscript.

We thank the reviewer for their positive comments.

*1. Both *Adgrd1* and *Plxdc2* are expressed in several tissues. How can the authors exclude the possibility that the *ADGRD1* (with *PLXDC2*) mediated signaling in the other tissues indirectly regulate the oviductal fluid production? Conditional KO or rescue by local gene delivery (eg., tissue-specific Tg, viral vectors, EP) would be best to answer. Alternatively, the authors may want to inhibit the pathway with antibody (or inhibitory domain) releasing beads, etc.*

We agree with the referee that establishing the role of gene effects locally would add to the manuscript. We have made a large number of attempts to try this, but have been frustrated by the technical difficulties involved; principally, these difficulties are centred on the fact that there is no good *in vitro* model of oviductal embryo transport.

As the reviewer suggests, the ideal way of doing this is through creating a conditional knockout, but this relies on having a suitable cre-driver line which enables allele recombination and inactivation exclusively within the cell types of interest. While we were able to find a cre-driver line that is expressed in the oviductal epithelium (*Wnt7a-Cre*¹) there is additional expression in other tissues both during the development of the female reproductive tissues (Mullerian duct) and in other epithelial cells (uterus) which would not unequivocally resolve this issue.

As the reviewer suggests, controlling gene activity locally might also provide answers to this question and among the experiments that we have tried to establish local gene effects are to apply the adenylyl cyclase inhibitor forskolin to short circuit the *Adgrd1* receptor by increasing intracellular cAMP levels² in the oviductal epithelium. Our prediction was that increasing the levels of intracellular cAMP would rescue the embryo transport block in *Adgrd1*-deficient oviducts. We first tried this by using explanted *Adgrd1*-deficient oviducts collected at 0.5 dpc and incubated overnight with 2 μ M, 10 μ M or 100 μ M forskolin. This approach did not rescue embryo transport, but because transport occurs over several hours, and explanted oviducts are separated from the blood circulation and nervous system, we were concerned that the integrity and function of the explanted tissue was compromised to too great a degree. We observed similar equivocal results when we used ligated wild-type oviducts collected at 0.5 dpc, 1.5 dpc and 2.5 dpc, and *Adgrd1*-deficient oviducts at 0.5 dpc to investigate the effects of both forskolin and a cell permeable cAMP analogue, dibutryl-cAMP on fluid secretion.

Other possibilities that we have investigated are exploring if *Adgrd1* function was related to reducing the flux of ions across the oviductal epithelium to regulate the passage of water into the lumen of the tube. To test this hypothesis, we used explanted ligated wild-type oviducts at 0.5 dpc and exposed them to 10 μ M bumetanide (a Na⁺:K⁺:2Cl⁻ co-transporter inhibitor) or 10 μ M 5-nitro-2-(3-phenylpropyl-amino) benzoic acid, (NPPB - a chloride channel blocker) for five hours, but both failed to show any effect on fluid accumulation. The experiment was repeated on wild-type oviducts at 0.5 dpc at higher doses of NPPB (both 100 μ M and 500 μ M). As NPPB is known to block the cystic fibrosis transmembrane conductance regulator (CFTR) which is expressed on the oviductal epithelium³, we repeated the treatment on two *Adgrd1*-deficient explanted oviducts at 0.5 dpc. The treated oviduct did not show any difference from the untreated contralateral oviduct. We further explored the possibility that by inducing smooth muscle relaxation, we might be able to recover the embryo transport defect, and so we therefore used explanted oviducts from two *Adgrd1*-deficient mice and two heterozygous controls and treated them with 0.2mM and 2mM NG-Monomethyl-L-arginine, monoacetate salt (L-NMMA) which is a NOS inhibitor but again we observed no effect. The overall conclusion was that because embryo transport required maintaining oviductal function for several hours, explanted oviducts were not suitable because their functional integrity was compromised once dissected from the animal.

To try to circumvent these limitations, we therefore tried to locally affect gene function in the oviduct within the living animal. One experiment that we attempted was modelled on the assay of placing glass beads into the oviduct which are also inappropriately retained in the ampulla of *Adgrd1*-deficient oviducts, as shown in our main manuscript. Here, we investigated the effect of forskolin *in vivo*, again in an attempt to bypass the *Adgrd1* receptor by immersing the beads in a medium containing 10 μ M forskolin. Again, this treatment did not prevent the embryo block at the AIJ even when repeated at a higher concentration (100 μ M) of forskolin. Again, we had concerns that it was not possible to control the amount of drug delivered *in vivo* within the oviduct. In particular, the rapid adovarian fluid flow is likely to have prevented forskolin from reaching the isthmus in sufficient concentration and time. Indeed, the rapid flow of fluid in the oviduct would prevent the local delivery of other reagents including other compounds, antibodies or viral particles.

Establishing if the genes act locally is something that we will continue to explore, and this may be aided by the development of better gene delivery systems and/or availability of more appropriate cre driver lines.

*2. The authors demonstrated that the *Adgrd1* is expressed in the part of the oviduct but did not show ovulated COCs. The reviewer would like to know if *Adgrd1* is expressed in COCs or not. If yes, the authors should show if *Adgrd1* KO COCs are functional in oviductal fluid regulation. Although it is not an easy experiment, this can be addressed by transplanting the KO ovary into WT females. The reciprocal experiments are welcome.*

We agree that this is an important experiment and we didn't make it clear that the data were present in the original submission. The X-gal staining which reports on *Adgrd1* promoter activity provided in the submitted manuscript was performed at ovulation on a female within an estrous cycle with COCs present in the ampulla. We have highlighted the COCs in an X-gal stained oviduct which, despite strong staining in the oviduct, shows no detectable *Adgrd1* promoter activity in the cumulus cells or oocyte in the COC (arrows - Fig. R1a). To be additionally confident that there is no *Adgrd1* expression in the COCs, we used the very sensitive technique of RT-PCR. Using cDNA prepared from both COCs and oviducts, we could show that *Adgrd1* is expressed in the oviduct but not COCs in contrast to an oocyte-restricted gene (*Juno*) which is expressed in COCs but not oviducts (Fig. R1b).

Fig R1: *Adgrd1* is not expressed in cumulus cells. (a) X-gal staining of an *Adgrd1*^{+/-} oviduct, shows reporter activity in the isthmus, the two arrows point to COCs that are visible inside the ampulla but exhibit no lacZ activity. (b) The expression of *Adgrd1* in COCs and oviducts obtained from wild type females was analysed by RT-PCR. The housekeeping gene *beta-actin* and the oocyte-specific gene *Juno* are used as controls. *Adgrd1* is undetectable in COCs and is expressed in the oviducts, while *Juno* is expressed exclusively in the oocytes.

3. *There is no apparent abnormality reported in Plxdc2 hypomorphic mutant mice* ([http://www.informatics.jax.org/marker/MGI\[informatics.jax.org\]:1914698](http://www.informatics.jax.org/marker/MGI[informatics.jax.org]:1914698)). *It should be addressed by cumulus cell-specific gene disruption (or complementation in whole KO), or at least discussed.*

In agreement with the reviewer's suggestion, we have added some text describing the hypomorphic mutant mice phenotype to the discussion in the manuscript as well as the possibility of redundancy with *Plxdc1*. Regarding the possibility of generating cumulus-cell specific conditional knock-outs, while this would be a good approach, to our knowledge, there are no identified genes that are exclusively expressed in the cumulus cells meaning there are no cumulus cell-specific transgenic cre-driver lines available. While there are examples of genes

that are expressed by granulosa cells, their expression is not restricted to this cell type. One example is the *Amhr2-Cre* allele where the Anti-Müllerian Hormone Receptor Type 2 gene drives the expression of the CRE recombinase in granulosa cells⁴; however, the promoter is active within the Müllerian duct from embryonic day 12.5 meaning that all the derived adult tissues which include the oviduct, uterine stroma and myometrium would be affected.

4. Does *PLXDC2* function in membrane form or secreted form?

The primary sequence of *PLXDC2* clearly contains a transmembrane-spanning region at the C-terminus of the polypeptide which suggests that it is a single-pass type I membrane protein and will therefore be localised within the plasma membrane. This is consistent with the cell surface staining we observe on cumulus cells with the anti-*PLXDC2* antibody (Fig. 5f, main manuscript), and also the Western blot performed on COCs which shows a single band corresponding to the molecular mass predicted for the full-length protein (Fig. 5e, main manuscript). Together, these data demonstrate that the majority of *PLXDC2* is present in cumulus cells as the full-length membrane-associated form, and we have no evidence to suggest that it might be cleaved from the cell surface or that a secreted isoform exists.

5. The readers may want to see the fallopian tube as extended in Figure 2a.

Agreed. This turned out to be more difficult than we had anticipated because the oviduct is extremely compact and highly coiled which meant that it had to be fixed before dissection and trying to extend the tissue; following fixation, however, the tissue became very brittle and had a tendency to tear. We were able to stretch out the oviduct sufficiently to make the expression of *Adgrd1* in the oviduct clearer and we were generally pleased with the results (Fig. R2). We have therefore substituted these new data for the original image in our revised manuscript.

Fig R2: Extending the fixed oviduct makes *Adgrd1* expression in the oviduct clearer. **a** *Adgrd1* promoter activity is high in the isthmus of the oviduct as detected by whole mount X-gal staining using the *lacZ* reporter enzyme. Staining is detected in the homozygous oviducts (left) but not in the wild-type sibling control (right). **b** Whole mount X-gal staining of homozygous *Adgrd1* oviducts that have been removed from the ovary and uterus and gently extended to show *Adgrd1* promoter activity in the isthmus.

Reviewer #2

This paper reports the findings of an investigation of the action of Adgrd1 in relation to mammalian reproduction. The transport of gametes through the fallopian tube and fertilization in the fallopian tube is a highly regulated time sensitive process necessary for reproduction. Infertility, subfertility and ectopic pregnancy can result when tubal transport is adversely affected. This reports the novel findings of Adgrd1 action by noting the response in deficient models, as well as localization within the oviduct. This is a impressive introductory work into this area. Future directions may include human fallopian tube specimens obtained from routine gynecologic surgeries (sterilization or hysterectomy). Very nice figure 4 to add to the discussion. Would like to see more in depth discussion of the clinical relevance in the discussion as opposed to a summary of the results, as well as discussion of limitations and future directions.

We thank the reviewer for their positive comments.

We have now extensively expanded the discussion in our revised manuscript to include a discussion on the clinical relevance of the findings as well as future directions. We have also briefly mentioned some of the experiments that we attempted (mentioned in response to reviewer 1's first point) which highlights some of the technical challenges which must be overcome to make further progress in this area.

Reviewer #3

The manuscript investigated mechanisms of how valve-like behaviors of ampullary-isthmic junction (AIJ) is controlled during the process of embryo transport in the oviduct. In the first part of the paper, it is clearly shown that the orphan adhesion G-protein coupled receptor Adgrd1 is required for the passage of embryos into the isthmus from the ampulla where fertilization occurs. In Adgrd1 mutant mice, embryos were arrested in the ampulla and reached the morula stage by E2.5. Transport of embryos through the oviduct is thought to be controlled by multiple factors including the unidirectional beating of cilia, contractions of muscles, and fluid secretion. The authors analyzed the effects of Adgrd1 mutation on those factors and proposed that dysregulation of fluid secretion in the Adgrd1 mutant oviduct caused the arrest of embryos in the ampulla. They finally explored molecules which activate Adgrd1 in the oviduct and found a candidate, Plxdc2.

Their finding that female mice lacking Adrg1 are sterile because embryos are arrested in the ampulla is potentially interesting. However, the manuscript is less convincing, when they tried to validate the role of the

fluid secretion on the passage of embryos through AIJ, which is the main point of the study. More direct evidence is needed to conclude that the control of oviductal fluid secretion by Adgrd1 is essential for unlocking the restraining mechanism at the AIJ. Furthermore, the effects of Adgrd1 mutation on cilia beating, muscle contraction, and tissue structures need to be more carefully and thoroughly analyzed. Those two major points require revisions.

Major points:

The most critical point is that the authors did not explain exactly why defects in fluid secretion are causative of the arrest of embryos in the Adgrd1 mutant oviduct. More direct pieces of evidence are needed to explain the mechanisms of unlocking the valve-like “tubal-locking”.

Changes in fluid secretion are at the base of the proposal of this paper. Yet, the data are not convincing enough; only a few examples are shown and the changes are not quantitatively and statistically assessed (Fig. 2 c-e and supplementary Fig.4).

We agree that observing the fluid flow in the oviduct rather than using fluid accumulation through ligation experiments would provide a more direct mechanism to explain the oviductal transport defect. The experimental challenges here are that this must be done *in situ* using delicate surgical procedures within a living animal and the oviduct is both very small and highly convoluted. We were encouraged by the recent publication from Hino and Yanagimachi⁵ where they surgically manipulated the oviduct into a chamber containing circulating buffer and injected a tracer dye (Indian ink) into the oviduct close to the uterus and observed the movement of the dye towards the ovary. Not only were their findings consistent with our oviductal ligation experiments in that fluid flow was reduced in wild type murine oviducts at 1.5 dpc, but this also suggested a possible method for visualizing the fluid flow directly in Adgrd1-deficient oviducts. To perform these experiments required the development of a heating jacket (described in new Supplementary Fig. 4) which kept the viscera at physiological temperature while the dye injections were performed during the surgical procedure. Using this approach, we observed that the behaviour of the tracer dye in oviducts of Adgrd1-deficient mice at 1.5 dpc was consistently different from the fertile controls. In the control animals (both wild type and heterozygous animals) the dye behaved similarly to that observed by Hino and Yanagimachi: the dye segregated into several boluses within the oviduct close to the uterus which then gradually moved towards the ovary in a manner that preserved the quantised nature of the dye distribution. By contrast, in the Adgrd1-deficient oviducts, the tracer dye rapidly dispersed along the entire length of the oviduct so that once filled, there was relatively little change in the distribution of the dye along the length of the oviduct over the course of the observation. These data show that in control oviducts, oviductal fluid flow is reduced and has a pulsatile character compared to the continuous, more rapid flow in the Adgrd1 knockouts. These observations are consistent with our interpretation that the post-ovulatory attenuation of oviductal fluid flow is

misregulated in *Adgrd1*-deficient mice causing the retention of the embryos within the ampulla leading to infertility.

We have added these new experiments to our revised manuscript as kymographs (Figure 4f) which provide a two-dimensional visualisation of oviductal fluid flow and show the difference between the control and mutant oviducts (shown below - Fig. R3).

Fig R3: Oviductal fluid flow is dysregulated in *Adgrd1*-deficient mice. Left panel: schematic showing the region of the oviduct used for producing the kymographs. Right panels: Kymographs showing the behavior of a tracer dye injected into the oviduct of *Adgrd1*-mutants compared to wild type littermate control. *Adgrd1*-mutant oviducts are rapidly filled with the dye which does not change over the time of observation whereas control oviducts segregate the dye into boluses which are gradually moved towards the ampulla.

We have also provided the raw data videos which we have arranged to show the control and *Adgrd1*-deficient oviducts side-by-side for direct comparison and to also synchronise the tracer dye injection time to facilitate comparisons (Supplementary Movie 3). To aid the reader, we have also provided annotated still frames from these videos which describe these effects which we have added to the revised manuscript as Supplementary Figure 5. Together, these data provide a direct visualisation of the fluid behaviour within the oviduct and support our hypothesis that oviductal fluid flow is responsible for the inappropriate retention of embryos within the ampulla.

We would like to reassure the reviewer that we did try to quantitatively assess the fluid accumulation phenotype using several approaches, but because of the highly convoluted nature of the oviduct, this wasn't possible to do with sufficient accuracy by measuring volumes from microscopic images. We attempted to quantify fluid volumes in ligated oviducts by injecting a rhodamine-dextran dye, but we found that these measurements, while showing encouraging trends, were not very reliable due to the large variability, possibly because we observed that the dyes had a tendency to adhere to the epithelial surface of the oviduct. We also tried to determine the mass of the oviducts before and after removal of the fluid but again found that the data generated using this approach were highly variable. The inherent variability in these

approaches would have meant using a very large number of mice to achieve statistical significance which was neither practical nor ethically permissible.

Minor points:

Supplementary Figure 3a

Sufficient details of muscle structure and organization are not given.

Supplementary Movie 1 and 2

Differences in the cilia activity and the muscle contraction between the wild type and the mutant oviduct should be quantitatively and statistically assessed.

Agreed. The data in Supp Fig 3a were provided to give the reader a whole-tissue perspective, and the movies showed the raw rather than processed data to demonstrate that there are no overt morphological differences between the wild type and mutant oviducts. Encouraged by the comments from this reviewer, we have formalised these data by quantifying our observations and providing statistical tests which we have re-organized into an entirely new main figure in our revised manuscript (new main Fig. 3); we believe that this has the additional benefit of improving the readability of the manuscript. In summary, we have measured the thickness of the myosalpinx in both the ampulla and isthmus, and find that there are no statistically significant differences between the wild type and controls (Fig. R4 below and Figure 3g in the main manuscript).

Figure R4. Quantification of the myosalpinx thickness of the ampullary and isthmic oviductal epithelium reveals no difference between *Adgrd1*-deficient mice and controls. The thickness of the myosalpinx is similar in controls (hues of grey) and *Adgrd1*^{-/-} (hues of red) in the different oviductal regions. Bars represent the mean \pm SEM, measurements were performed on 3 *Adgrd*^{+/+} and 3 *Adgrd1*^{-/-} oviducts; a minimum of two sections per mouse were analysed.

We have also stained sections of the oviduct with a smooth muscle marker which shows no difference between mutant and controls and is shown in our revised manuscript (Fig. R5 below and Figure 3f and f' in the main manuscript)).

Figure R5. Staining sections of the isthmus with a smooth muscle marker reveals no difference between *Adgrd1*-deficient mice and controls. Representative examples of sections of the isthmus stained with anti-smooth muscle alpha actin (magenta) and counterstained with DAPI (Cyan). Oviducts from fertile heterozygous controls are shown in the left panel and infertile *Adgrd1* oviducts on the right.

We have also extended this analysis to examine the secretory cells within the oviduct. In brief, we analysed the oviducts of age-matched wild type and *Adgrd1*-mutant mice in diestrous by staining them with a secretory cell marker (PAX8) and found that there were no differences in the number or organization of secretory cells (Fig. R6 below and Supplemental Figure 3 in the manuscript).

Figure R6. The relative number and distribution of secretory cells appears normal in *Adgrd1*-deficient oviducts. **a**, No overt defect is visible in the distribution of secretory cells in the epithelium of control *Adgrd1*^{+/-} and *Adgrd1*^{-/-} oviducts. Oviductal sections from adult females in diestrous were stained with an antibody against PAX8 (green), and nuclei counterstained with DAPI (blue, merged with green in the right panels). **b**, The number of secretory cells positively stained for PAX8 was counted in a minimum of two sections per each animal in both the ampulla and the isthmus. Each shade of color represents a female oviduct, controls are in shades of gray and KOs are in shades of red. The percentage of secretory cells is higher in the isthmus than in the ampulla in both controls and *Adgrd1*^{-/-} oviducts. A two-way ANOVA analysis found that the genotype has no effect while the difference in percentage of secretory cells between the ampullary and isthmic regions of the oviduct is extremely significant ($p < 0.0001$), as expected.

Finally, to quantify ciliary function, we have measured the velocity of beads placed on explants of the ampullary epithelium. These data (Figure R7 below and Figure 3d in the main manuscript), and again we find no difference between the wild type and mutant tissue.

Figure R7. Polystyrene beads placed on oviductal epithelium explants moved at equivalent speeds in an abovarial direction in both control and *Adgrd1*^{-/-} epithelial tissues. Individual data points are bead velocity quantified with Image J manual tracking for a minimum of 15 seconds. Bars represent the mean \pm SEM, measurements were performed on 3 *Adgrd*^{+/+} and 3 *Adgrd1*^{-/-} ampullae. An unpaired t-test analysis showed no significant difference between the groups.

Rebuttal references

1. Winuthayanon, W., Hewitt, S. C., Orvis, G. D., Behringer, R. R. & Korach, K. S. Uterine epithelial estrogen receptor α is dispensable for proliferation but essential for complete biological and biochemical responses. *Proc. Natl. Acad. Sci. U. S. A.* **107**, 19272–19277 (2010).
2. Liebscher, I. *et al.* A tethered agonist within the ectodomain activates the adhesion G protein-coupled receptors GPR126 and GPR133. *Cell Rep.* **9**, 2018–2026 (2014).
3. Chen, M. *et al.* Functional expression of cystic fibrosis transmembrane conductance regulator in rat oviduct epithelium. *Acta Biochim. Biophys. Sin.* **40**, 864–872 (2008).
4. Robker, R. L. *et al.* Identification of sites of STAT3 action in the female reproductive tract through conditional gene deletion. *PLoS One* **9**, e101182 (2014).

5. Hino, T. & Yanagimachi, R. Active peristaltic movements and fluid production of the mouse oviduct: their roles in fluid and sperm transport and fertilization†. *Biol. Reprod.* **101**, 40–49 (2019).

Reviewers' Comments:

Reviewer #1:

Remarks to the Author:

The authors sufficiently addressed the points I raised. The reviewer agrees that some experiments are challenging with currently available genetic tools. The reviewer really appreciates the authors' sincere effort during such a difficult time.

Reviewer #3:

Remarks to the Author:

The authors carried out additional experiments which are not easy, and thanks to their efforts the manuscript is improved very much. The revised manuscript provides novel findings and is satisfactory and now the paper should be accepted for publication.

Response to reviewers' comments:

Reviewer #1 (Remarks to the Author):

The authors sufficiently addressed the points I raised. The reviewer agrees that some experiments are challenging with currently available genetic tools. The reviewer really appreciates the authors' sincere effort during such a difficult time.

Reviewer #3 (Remarks to the Author):

The authors carried out additional experiments which are not easy, and thanks to their efforts the manuscript is improved very much. The revised manuscript provides novel findings and is satisfactory and now the paper should be accepted for publication.

We thank the reviewers for their comments.